# Is the Novel Slot Blot a Useful Method for Quantification of Intracellular Advanced Glycation End-Products?

**DOI:** 10.3390/metabo13040564

**Published:** 2023-04-16

**Authors:** Takanobu Takata

**Affiliations:** Division of Molecular and Genetic Biology, Department of Life Science, Medical Research Institute, Kanazawa Medical University, Uchinada 920-0293, Ishikawa, Japan; takajjjj@kanazawa-med.ac.jp; Tel.: +81-76-286-2211

**Keywords:** advanced glycation end products, slot blot, Tris, urea, thiourea, 3-[(3-cholamidopropyl)-dimethyl-ammonio]-1-propane sulfonate, polyvinylidene difluoride membrane, enzyme-linked immunosorbent assay, gas chromatography–mass spectrometry, liquid chromatography–electrospray ionization–mass spectrometry

## Abstract

Various types of advanced glycation end-products (AGEs) have been identified and studied. I have reported a novel slot blot analysis to quantify two types of AGEs, glyceraldehyde-derived AGEs, also called toxic AGEs (TAGE), and 1,5-anhydro-D-fructose AGEs. The traditional slot blot method has been used for the detection and quantification of RNA, DNA, and proteins since around 1980 and is one of the more commonly used analog technologies to date. However, the novel slot blot analysis has been used to quantify AGEs from 2017 to 2022. Its characteristics include (i) use of a lysis buffer containing tris-(hydroxymethyl)-aminomethane, urea, thiourea, and 3-[3-(cholamidopropyl)-dimetyl-ammonio]-1-propane sulfonate (a lysis buffer with a composition similar to that used in two-dimensional gel electrophoresis-based proteomics analysis); (ii) probing of AGE-modified bovine serum albumin (e.g., standard AGE aliquots); and (iii) use of polyvinylidene difluoride membranes. In this review, the previously used quantification methods of slot blot, western blot, immunostaining, enzyme-linked immunosorbent assay, gas chromatography–mass spectrometry (MS), matrix-associated laser desorption/ionization–MS, and liquid chromatography–electrospray ionization–MS are described. Lastly, the advantages and disadvantages of the novel slot blot compared to the above methods are discussed.

## 1. Introduction

During the last century, several advanced glycation end-products (AGEs) have been identified and studied [1,2]. Many AGEs are produced in certain organs and tissues and circulate in the body through the bloodstream, whereas dietary AGEs are taken up by the body through food and drink consumption [1,2]. Researchers have measured various AGEs in the blood and analyzed their relationships with many diseases, particularly lifestyle-related diseases (e.g., diabetes, cardiovascular disease). In fact, some AGEs in the blood may serve as biomarkers for some diseases [1,2]. As dietary AGEs are taken into the body and are known to cause health effects through receptors for AGEs, they have been extensively studied and quantified by researchers [1,2]. On the contrary, the quantification of intracellular AGEs in some organs during clinical investigations has proven challenging. However, their quantification in vitro or in animal models has been conducted to investigate their impacts on the regulation of certain components (e.g., proteins and reactive oxygen species) in diseased states (e.g., hyperglycemia and diabetes) [3,4,5,6].

Several studies have quantified AGEs using various methods, such as western blotting [3,4], immunostaining [5,6,7], enzyme-linked immunosorbent assay (ELISA) [5,8,9], gas chromatography–mass spectrometry (MS) (GC–MS) [10], matrix-associated laser desorption/ionization-MS (MALDI–MS) [11], liquid chromatography–electrospray ionization–MS analysis (LC–ESI–MS) [12], and slot blot analysis [13,14,15,16]. The data obtained in these studies were analyzed using statistics to evaluate significant increases or decreases in AGE levels.

The slot blot analysis is a useful technology because it can detect AGEs through the use of anti-AGE antibodies, which can be easily obtained or prepared. Furthermore, the analytical samples do not require any pretreatment, such as acid hydrolysis, derivatization, or enzymatic digestion. However, the structure of individual AGEs cannot be identified through this technique.

When using the slot blot method, the intracellular content of AGEs can be quantified and calculated, especially for standard AGEs (e.g., AGE-modified bovine serum albumin (BSA)); however, the statistical analysis of the data had not been reported until 2017. Interestingly, in 2015, Koriyama et al. performed semi-quantification of the intracellular content of one of glyceraldehyde-derived AGEs (GA-AGEs, also termed as toxic AGEs (TAGE)) using slot blot analysis with a radioimmunoprecipitation (RIPA) buffer and nitrocellulose membranes; although statistical analysis was carried out, the value for standard AGEs could not be determined [14]. In contrast, in 2017, we developed a novel slot blot analysis method that could calculate the amount of intracellular TAGE in the human pancreatic ductal cell line PANC-1 based on a standard TAGE-modified BSA (TAGE-BSA) [15]. To this end, we selected a polyvinylidene difluoride (PVDF) membrane, which exhibits stronger absorption of proteins than the nitrocellulose membrane. However, the proteins dissolved in RIPA buffer were inefficiently adsorbed onto the PVDF membrane. Therefore, we utilized a more suitable lysis buffer that contained tris-(hydroxymethyl)-aminomethane (Tris), urea, thiourea, and (3-[(3-cholamidopropyl)-dimethyl-ammonio]-1-propane sulfonate) (CHAPS) [15]. This method is novel in that it uses a lysis buffer with a composition similar to that used in two-dimensional gel electrophoresis (2DE)-based proteomic analyses [17,18,19].

From 2017 to 2022, we quantified the intracellular content of TAGE, which was calculated using TAGE-BSA in the human hepatic cell line HepG2 [20,21], hepatocyte-like cells differentiated from human induced pluripotent stem cells (hiPS-HLC) [22], rat primary hepatocytes [23], rat primary cardiomyocytes [24], human normal cardiac fibroblasts [25], the human pancreatic islet β-cell line 1.4E7 [26], a human pancreatic ductal cell line [15], the murine skeletal muscle myoblast cell line C2C12 [27], the murine osteoblastic cell line MC3T3-E1 [28], and rat hepatic tissue [23,29]. In addition, in 2019, we suggested that human hepatocytes might produce 1,5-anhydro-D-fructose-derived AGEs (1,5-AF-AGEs) [30]. Although these are generated by 1,5-AF or combined with proteins, their presence in cells and blood remains unclear. In 2019, we succeeded in quantifying intracellular 1,5-AF-AGEs in HepG2, which were treated with 1,5-AF, with 1,5-AF-AGE-modified BSA as a standard, using the novel slot blot analysis [30]. Although slot blotting is an analog technology that has been used to detect and quantify RNA [31,32,33], DNA [34,35], and proteins [36,37,38,39,40] since around 1980, the novel slot blot analysis is currently the ideal method for AGE quantification. In this review, the quantification of AGEs using our developed novel slot blot as well as other methods (traditional slot blot, western blot, immunostaining, ELISA, GC–MS, MALDI–MS, and LC–ESI–MS) were compared. Further, the advantages and disadvantages of each were compared.

## 2. Traditional Slot Blot for Detection/Quantification of RNA, DNA, and Proteins

Various researchers have detected and quantified RNA [31,32,33], DNA [34,35], and proteins [36,37,38,39,40] since around 1980. Notably, the components of the extraction buffers for RNA [31,32,33] and DNA [34,35] detection are different from those for proteins. Furthermore, their data have not been analyzed using statistics. In contrast, proteins have been analyzed using slot blotting to detect and quantify RNA and DNA [36,37,38,39,40]. Many researchers have selected nitrocellulose membranes but not PVDF membranes [36,37,38,39,40], although the absorption ability of PVDF membranes is comparatively higher. The reason may be that the components of cell lysates inhibit the samples probed onto PVDF membranes.

In the slot blot analysis, quantification of the target proteins, which is calculated using a standard protein, is not performed. Although Ghiani et al. performed semi-quantification of allergens, they calculated the integrated optical density (IOD)/standard allergen IOD, and only used one lane of the standard allergen for normalization [37]. They did not draw a standard curve of the allergen for quantification. Kumar et al. and Grabias et al. quantified the recombinant Plasmodium falciparum circumsporozotie protein (rPfCSP) and native PfCSP from oocysts (PfOocys) [38,39]. In their research, the degree of reliability of slot blot was compared with the western blot, which also blotted the same amount of rPfCSP (1.25 pg) [38]. However, they detected PfOocys using slot blot, which were exposed to a digital scan and film and were quantified in the cell lysate [39]. Nimmo et al. detected and quantified α-synuclein, which was probed onto a nitrocellulose membrane within a range of 20–200 ng (four points) [40]. Although they performed quantification, they did not perform statistical analyses, and the error bar was not shown in the graph. This investigation may have been performed on the basis of the antibody used to probe α-synuclein.

## 3. Methodology and Characteristics of the Novel Slot Blot

### 3.1. Equipment

The novel slot blot method was performed using a Bio-Dot SF Microfiltration Apparatus (Cat. No.: 170-6452, Bio-Rad Laboratories Inc., Hercules, CA, USA) (slot blot apparatus with 48 lanes) (Figure 1).

### 3.2. Preparation and Application of Lysis Buffer

The components of the lysis buffer were modified with a buffer prepared for 2DE-based proteomic analyses. First, Tris (Cat. No.: 011-20095; Fujifilm Wako Pure Chemical, Osaka, Japan), urea (Cat. No.: 217-01215; Fujifilm Wako Pure Chemical), thiourea (Cat. No.: 201-17355; Fujifilm Wako Pure Chemical), and CHAPS (Cat. no.: 347-04723; DOJINDO Laboratories, Kumamoto, Japan) were dissolved in ultra-pure water to make a solution of 30 mM Tris, 7 M urea, 2 M thiourea, and 4% CHAPS (Solution A, Table 1). Second, a protease inhibitor cocktail tablet (Complete Tablets EDTA-Free, EASY pack, Cat. no.: 04-693-132-001; Roche, Penzberg, Bavaria, Germany) was dissolved in ultra-pure water (final volume: 2 mL, Solution B, Table 1 [15,24]). Finally, Solution A and Solution B were mixed together (9:1) (Solution C, Table 1). The components of Solution C were 27 mM Tris, 6.3 M urea, 1.8 M thiourea, and 3.6% CHAPS, making up the lysis buffer used for our novel slot blot analysis [15,22,23,24,25,26,27,29]. The final components of Solution D were based on the preparation of Solution C. When Tris, urea, thiourea, and CHAPS were dissolved in ultra-pure water, Solution B was added to it (Table 1) [20,21,28,30].

The number of references used for preparing Solution C was 8 and that for Solution D was 4, for the quantification of intracellular AGEs using the novel slot blot (Table 2). Although there are no studies in which each solution was used against the same cells (Table 3 and Table 4), I considered both, Solution C and D, to be suitable for preparing the cell lysate and, therefore, used them to perform the novel slot blot analysis.

Based on the preliminary examination, I hypothesized that the RIPA buffer components might cause denaturation and inhibition of the probing of proteins onto the PVDF membrane. Consequently, owing to its ability to denature proteins, RIPA buffer might be useful in western blotting assays where the components are removed during electrophoresis but not in slot blot analyses. Therefore, I concluded that a suitable cell lysate solution would denature proteins without inhibiting their probing onto the PVDF membrane. Based on these criteria, I chose a lysis buffer containing Tris, urea, thiourea, and CHAPS [15]. Interestingly, lysis buffers with the same composition but different reagent concentrations have been used in 2DE analyses [17,18,19]. Many of these studies used 7 M urea and 2 M thiourea [17,18,19,41,42,43,44,45,46,47]. However, the concentration of CHAPS was kept at 2% [42,44], 3% [17,47], or 4% [18,19,42,43,45,46]. I decided to use 4% CHAPS in Solution A, which was then 0.9-fold diluted to prepare Solution C (Table 1) [15,24]. Further, Solution D was the modified solution based on Solution C and was selected for our slot blot (Table 1) [30].

McCathy et al. and Herbert suggested that urea, thiourea, and CHAPS can denature proteins by acting as chaotropic reagents and surfactants [41,42]. Chaotropic reagents, such as urea and thiourea, disrupt hydrogen bonding and cause protein unfolding, thereby exposing hydrophobic amino acid residues to the solution [41,42]. Urea is the most commonly used chaotropic reagent in 2DE analyses, whereas thiourea/urea combinations are broadly used to exploit the improved denaturing ability of thiourea [41]. In addition, CHAPS is used in combination with urea and thiourea to coat hydrophobic residues and improve solubility [41,42]. Furthermore, Tris has been used to stabilize the pH range of cell lysates at 8.5–8.8 [17,19]. Whereas previous studies have used 30 mM [18,19] or 40 mM [17] of Tris, I used 3.6 mM or 4.0 mM as the final Tris concentration in the lysis buffer with a pH of 8.5 (Solution C and Solution D, Table 1). Although there are no data for the differences in slot blot analysis between Solution C and D, the background absorption of the components of Solution C was found to be lower than that for Solution D when the protein concentration of samples was measured using the Bradford method. Furthermore, Solution C contained 2.5-fold components of the protein inhibitor compared to the standard protocol (Roche Applied Science, Penzberg, Germany) (Table 1). Therefore, Solution C may be more suitable when the protein concentration of the sample cell lysates is predicted to be low.

The core technology included a lysis buffer containing Tris, urea, thiourea, and CHAPS [15], although the reagent concentrations may be modified. Interestingly, there have been four reports of modified methods in which cell lysates and standard TAGE-BSA were dissolved in Solution D containing protease inhibitors, based on the preparation method for Solution C, similar to this protocol [20,21,28,30]. We first reported this novel slot blot analysis with Solution C in 2017 [15], whereas Papadaki et al. reported the semi-quantification of methylglyoxal-derived AGEs (MGO-AGEs) using a dot blot method (similar to slot blot) with urea and sodium dodecyl sulfate (SDS) in 2018 [16]. Human left ventricle tissues were homogenized and the obtained proteins were dissolved in standard rigor buffer supplemented with 1% Triton and phosphatase inhibitors. After washing out Triton and centrifuging, the pellet was resuspended in 8 M urea and 1% SDS [16]. Because they used 8–9 M urea for sample preparation, their samples were analyzed using 2DE [42,48,49,50], which provided scope to test the potential uses of reagents applied in 2DE examination. Some studies that performed 2DE analyses did not use Tris [44]; rather dithiothreitol [45], Triton-X100 [46], and formic acid [47] were used. Although the lysis buffer might be improved with the use of other reagents, I hypothesized that Solution C and D were more suitable for the preparation of cell lysates and slot blot analysis. The cell lysates were prepared as previously described [15,20,21,22,23,24,25,26,27,28,29,30].

### 3.3. Application of Novel Slot Blot Assay for Standard AGE-Modified Protein Aliquots, Horseradish Peroxidase (HRP) Marker Solution, and Sample Solution

Based on the manufacturing protocol of Bio-Rad, phosphate-buffered saline (PBS)(−) and other solutions were probed onto a PVDF membrane under vacuum in a water aspirator. Figure 2 depicts the model image of application of the novel slot blot assay for a standard AGE-modified protein solution, HRP marker (Cat. no.: BNP-M41; Bionexus, Oakland, CA, USA) solution, and sample solution. I quantified intracellular TAGE in C2C12 cells in which a rabbit polyclonal anti-TAGE antibody as well as TAGE-BSA (obtained from Prof. Takeuchi of the Department of Advanced Medicine, Medical Research Institute, Kanazawa Medical University, Uchinada, Japan) were used [27,51].

In the general slot blot analysis of the intracellular AGE contents, the standard AGE-modified protein aliquots, HRP marker solution, and sample solution were applied to the PVDF membrane with *n* = 1 or 2 (Figure 2). The general slot blot apparatus had 48 lanes (Figure 1). The target proteins were separately probed with regular and neutralized anti-AGE antibodies as they exhibited non-specific binding when probed together with other antibodies. Although the standard TAGE-BSA aliquots were singly applied to the PVDF membrane (*n* = 1), experimental samples were applied in duplicate (*n* = 2) when it was possible with the number of available lanes [26,27]. I also performed more than three independent experiments for the slot blot analysis and statistically analyzed the data [15,20,21,22,23,24,25,26,27,28,29,30]. In addition, as treatment with neutralized anti-AGEs antibodies was not needed, regular anti-AGE antibodies were used [20,21,28].

### 3.4. Chemiluminescene Imaging of Bands on Membranes and Evaluation of AGE Content

Incubation with anti-AGEs antibody (e.g., anti-TAGE antibody) and secondary antibody (Cat. no.: P0448; Dako, Glostrup, Denmark) solution and washing of PVDF membranes were performed as previously described [15,22,23,24,25,26,27,29]. After adding chemiluminescent kit reagents (ImmunoStar LD, Cat. no.: 292-69903; Fujifilm Wako Pure Chemical), the PVDF membrane was incubated at room temperature for the same period. Each PVDF membrane treated with the respective pair of antibodies (regular and neutralized) was simultaneously visualized using a Chemilumi imager (Cat. no.: Fusion FX; M&S Instruments Inc., Osaka, Japan). As shown in Figure 3, anti-TAGE antibody was probed on the left side of the developed PVDF membrane, whereas the neutralized anti-TAGE antibody was probed on the right side. The HRP marker solution was used as a control. The corrected luminance value (arbitrary unit, AU) was calculated using the formula mentioned below.

Calibration-corrected luminance value = (HRP^Left^ − Blank^Left^)/(HRP^Right^ − Blank^Right^).

Difference in luminance value between the one in the left side and the corrected one in the right side of the PVDF membrane (AU) = (A^Left^ − Blank^Left^) − (A^Right^ − Blank^Right^) × calibration-corrected luminance value.

In this examination, only the component (A^Right^ − Blank^Right^) was multiplied with the calibration-corrected luminance value, which was used for the bands of HRP marker solution on both right and left sides of the PVDF membrane.

The standard curve was drawn based on the corrected luminance values (AU) obtained from standard AGE (AGE-modified protein) aliquots. The AGE content in each sample was then calculated based on this standard curve. However, we did not determine the luminance value of the loading control protein (β-actin and β-tubulin) in the final step [15,24]. This is because we considered that (i) the slot blot analysis should be performed similar to ELISA, and (ii) anti-loading control protein antibodies for the sample are unable to recognize the standard AGE (e.g., TAGE-BSA [27]) aliquots in the same lane. Furthermore, when staining the samples on the PVDF membrane with Coomassie brilliant blue, the bands of standard AGE aliquots are not stained if the protein content range (0–100 ng) is below the detection limit.

## 4. Comparison between the Novel Slot Blot Method and Other Methods for AGEs Quantification

### 4.1. Comparison with the Traditional Slot Blot or Dot Blot

TAGE and *N*^ε^-carboxymethy-lysine (CML) were semi-quantified, and the data were statistically analyzed [14,52,53]. Because the value was calculated as the luminance of AGEs per β-actin, this method can be normalized with a loading control (i.e., β-actin). Although the scattering of the absorption of AGEs may be calibrated, the AGE content per sample (e.g., AGE mass per protein) cannot be analyzed. Therefore, researchers cannot compare the AGE content of samples with that calculated using other methods. In addition, nitrocellulose membranes were selected for the other experiments, because PVDF membranes may have been unsuitable for use in slot blotting [14,52,53], although some studies still report their use [13,54]. Bronowicka-Szydełko et al. blotted serum samples onto PVDF membranes and used anti-MGO-derived AGEs as primary and secondary antibodies or only secondary antibodies for probing the proteins [13]. Furthermore, they quantified the luminance value of the MGO-AGEs per band of the secondary antibody. They also removed the effects of nonspecific combinations on the samples. This methodology was unique and may be beneficial for slot blot analysis. However, the resultant band images on the PVDF were very pale [13]. While the reason for this remains unclear, it is possible that the absorption of MGO-AGEs on PVDF membranes may be lower than that on nitrocellulose membranes.

In 2010, Takino et al. quantified intracellular TAGE content by using TAGE-BSA with a PVDF membrane [54]. Their method contained two important steps: (i) the probing of TAGE-BSA, HRP marker, and sample solution onto a PVDF membrane, and (ii) the lack of nonspecific binding with the tested antibodies. The PVDF membrane blotted with standard AGEs, HRP marker, and sample solution was cut into two smaller membranes. Each membrane was incubated with regular or neutralized anti-AGE antibodies. Following chemiluminescence detection of the protein bands, the luminance values of the bands on each membrane were calculated and corrected against that of an HRP marker, which was also blotted on each membrane. Because their strategy was useful, it was adopted for the novel slot blot method in this review. However, Takino et al. did not statistically analyze the value of intracellular TAGE [54]. As the error bar was not shown on the graph, the data might not have been the average of the values quantified with multiple examinations. As they selected RIPA buffer and PVDF membranes for slot blot analysis as well as western blot analysis, their aim may have been to investigate the tendency of intracellular TAGE levels to increase, using both slot blot and western blot. Their results were expressed as arbitrary units (U): 1 U corresponded to 1 μg of TAGE-BSA, and 1 mU corresponded to 1 ng of TAGE-BSA. Furthermore, they expressed the value (mU) in the graph of intracellular TAGE per 30 μg of protein, because they applied 30 μg of protein to the PVDF membrane. Therefore, I converted the values of their data to a single unit, that is, μg/mg protein (Table 5). When cells were treated with glyceraldehyde, cell viability decreased and intracellular TAGE increased [15,22,24,25,26,27,54]. Although this tendency was proven by both slot blot analyses, the value that Takino et al. quantified with their slot blot was very low compared to that of the novel slot blot (Table 5). This might be owed to the compositional differences in each lysis buffer.

Although Papadaki et al. used a lysis buffer containing 8 M urea and 1% SDS for dot blotting, which quantified MGO-AGEs in the left ventricular myofilament of patients with diabetes and heart failure (dbHF) in 2018 [16], they did not perform an analysis based on the standard MGO-modified protein. Semi-quantification was performed, in which the luminance of MGO-AGEs was normalized to that of β-tubulin. Because the components of their lysis buffer were similar to those of our novel slot blot, the proteins might be strongly probed onto the membrane. However, they selected the nitrocellulose membrane, not the PVDF membrane, to perform dot blotting and not slot blotting. It is possible that the PVDF membrane was unsuitable for dot blotting or slot blotting. Furthermore, they selected their dot blot to investigate the tendency of MGO-AGEs in the dbHF group to increase compared to those in the non-failing control group. They succeeded in the detection of MGO-modified peptides and identification of the MGO-modified site of the amino acid sequence in actin and myosin [16].

For the comparison of certain factors, such as cell viability vs. generation/accumulation of AGEs, semi-quantification of the traditional and novel slot blots may be beneficial. However, the novel method may be more convenient when comparing results from different experiments, because it can quantify AGEs based on calculations with standard AGEs. The novel slot blot and the one developed by Takino et al. have a similar conceptual design. Therefore, neither method was superior or inferior. However, the amount of sample quantified in vitro in the novel slot blot was 2 μg [15,22,24,25,26,27]. Therefore, the detection efficiency of our novel slot blot method may be higher than that of the method proposed by Takino et al. (Table 5) [54].

### 4.2. Comparison with Western Blot

Western blot analysis reveals the molecular weight of proteins in samples [15,54,55]. However, the mass of each protein remains unclear; hence, many researchers resort to semi-quantifying the target protein instead. In one study, when western blot analysis was used, the target proteins were detected with antibodies, and housekeeping proteins, such as β-actin, β-tubulin, and glyceraldehyde-3-phosphate dehydrogenase, were detected with their antibodies, because they were used as the loading controls. The value of the target protein in the treatment group was then compared to that of the control group, and semi-quantification was performed [55]. The luminance values of the target proteins were normalized to those of the loading control protein [15,54,55]. One band was detected in the western blot image. However, various bands were detected in the images of AGE-modified proteins (Figure 4) [3,4,15,55]. Therefore, if researchers quantify AGEs with western blotting, they should ensure that (i) total bands of AGE-modified proteins are targeted, and (ii) normalization with the loading control is performed. CML, a type of AGE, can modify various proteins [3]. Mastrocola et al. quantified CML- and *N*^ε^-(carboxyethyl)-lysine (CEL)-modified proteins, which were detected between the high molecular and low molecular weight areas of membranes [3]. In contrast, Chang et al. incubated the murine cardiomyocyte cell line HL-1 with high glucose, and a band of 40 kDa was detected and quantified [4]. Although they did not describe the type of AGE quantified, the anti-AGE antibody was obtained from Trans Genic (Fukuoka, Japan). I considered that semi-quantification based on the western blot image of AGEs requires more processing steps and time than the slot blot method. Moreover, this concept was very similar to the semi-quantification of AGEs as performed in the traditional slot blot. The novel slot blot analysis, however, can quantify AGEs in the samples (e.g., cell lysate, tissue lysate), which are calculated using standard AGEs for western blot [15,20,21,22,23,24,25,26,27,28,29,30]. Unfortunately, the molecular weight of each AGE-modified protein cannot be determined. For that purpose, western blot analysis would prove more useful than the novel slot blot analysis.

### 4.3. Comparison with Immunostaining

Immunostaining of cells in vitro [15,24] and tissues in vivo [56,57,58,59,60] is a beneficial visualization technique. In particular, the immunostaining of tissues can provide various types of information. Researchers can perform staining with some methods simultaneously (e.g., hematoxylin and eosin [56,58], DAPI reagent [56,57], oil red O [23,59,60], Sirius red [23], anti-interleukin antibody [29], and anti-AGEs antibody [5,6,7,23]) (Figure 5). Therefore, the accumulation of AGEs and conditions (e.g., steatosis and fibrosis) in tissues can be simultaneously analyzed. Jung et al. stained rat salivary glands with an anti-AGE antibody (Cat. no.: 6D12; Trans Genic, Kobe, Japan) and quantified AGEs using signal intensity (AU) [5]. Because the images of the areas were the same, the total signal intensity was quantified. Using the same methods, argpyrimidine and pentosidine in pancreatic islets of obese mice were stained [6]. The positive areas of anti-argpyrimidine and pentosidine antibodies were probed on the tissue per islet and then measured. The visual information of the islets was assessed simultaneously.

Further, LeWinter et al. investigated the location of CML in the cardiac tissue of patients with coronary bypass grafting; moreover, they quantified the CML count per μm^2^ in the cytoplasm and mitochondria in the cardiac tissue of brain-dead patients [7]. Although quantification via immunostaining of tissues with anti-AGE antibody could be associated with other examinations, absolute quantification and quantification using the standard AGE-modified protein could not be performed. Slot blot analysis can quantify AGE content in the samples (e.g., cell lysate, tissue lysate), which is calculated using standard AGEs [15,20,21,22,23,24,25,26,27,28,29,30]. This would be impossible in an immunostaining assessment [5,6,7,23]. However, the relationship between the accumulation of AGEs and condition of the tissue could be visualized through immunostaining but not through the novel slot blot. This is an advantage of immunostaining that is not present in other experimental methods.

### 4.4. Comparison with ELISA

ELISA can quantify targets that are calculated using standards similar to slot blot analysis. Furthermore, many commercial ELISA kits can be used to analyze major proteins. ELISA can be performed on various samples such as serum [5,61,62,63,64,65,66,67], plasma [68], urine [69], saliva [5], supernatants [9,62], cell lysates [8,70], and tissue lysates [8]. The target in ELISA is generally one protein, such as interleukin (IL)-1β, IL-6, IL-17, IL-18, or tumor necrosis factor (TNF)-α [61,62,70]. In contrast, the analysis and quantification of AGEs, which were performed with ELISA, showed greater diversity. The type of ELISA measurement of AGEs can be divided into (i) one free type of AGEs (e.g., pentosidine) or one type of AGE-modified protein (e.g., CML-modified human serum albumin (HSA)), and (ii) crude AGE-modified proteins. Interestingly, pentosidine, a free AGE, was measured using a novel competitive ELISA by Kashiwabara et al. [69]. They used biotin-conjugated pentosidine to coat each well of 96-well microplates and quantified it in urine samples. Because they used the standard pentosidine solution and were able to draw the standard curve, absolute quantification of pentosidine was performed. Therefore, they reported the results of pentosidine measurement in urine, which was in pmol/mL. Kinoshita et al. prepared *N*^ω^-(carboxymethyl)-arginine (CMA)-modified type I collagen, CMA-modified HSA, CML-modified type I collagen, and CML-modified HSA and subsequently measured them [71]. Yuki et al. analyzed the inhibition of CMA- and CML-modified gelatin from ribose and gelatin with the extracts of natural products using ELISA [72]. Ban et al. prepared *N*^δ^-(5-hydro-5-methyl-1,4-imidazolone-2yl)-ornithine (MG-H1)-modified gelatin and analyzed its inhibition with the extracts of natural products similar to Yuki et al. [73]. These ELISA results were assessed by measuring the absorbance (492 nm) [70,71,72].

Although AGEs have fluorescence, Indyk et al. and Pinto et al. especially focused on the fluorescence of pentosidine and its measurement in serum and plasma; the excitation/emission wavelengths were set at 335/385 nm (AU) [63,68]. In other studies, crude AGE proteins (e.g., various MGO-AGE-modified proteins) in serum [63,65,66,67], plasma [68], saliva [5], cell lysates [9], and tissue lysates [8] were measured. As the fluorescence of AGEs can be measured at an excitation wavelength of 370 nm and an emission wavelength at 440 nm, Indyk et al. and Pinto et al. measured the total AGEs in serum and plasma (AU) [63]. However, some reports have quantified AGEs using standard AGE-modified proteins, e.g., low-molecular weight fraction of melibiose-derived glycation products [63], GA-AGE-BSA [65], TAGE-BSA [66], 1,5-AF-AGE-modified BSA [67], CML-modified BSA [8], and MGO-modified BSA [9]. Because researchers have quantified the mass of AGE-modified proteins as a standard, the amounts of crude AGEs have been calculated with their AGE-modified proteins. If MGO-BSA and anti-MGO-AGE antibodies are used, they can quantify the amounts of MGO-AGEs in the cell lysates, which may contain MGO-modified heat shock protein (HSP)27, MGO-modified HSP70, and MGO-modified HSP90.

The novel slot blot method is similar to ELISA, as it quantifies AGEs with standard AGE-modified proteins. Similarities include the following: (i) the target is crude AGEs in samples, (ii) standard AGE-modified protein is needed, and (iii) loading control against each sample does not exist, although researchers can use it in western blotting. Therefore, a comparison between the novel slot blot method and ELISA is important. Because researchers need only anti-AGEs antibodies and standard AGEs to perform slot blot analysis, it is more convenient than ELISA. In contrast, the commercial ELISA kit that researchers use to quantify AGEs may be unavailable, and the laboratory-made ELISA by researchers requires cumbersome operations. These results suggest that the novel slot blot analysis is better than ELISA. However, the novel slot blot method can measure only two types of AGEs (TAGE and 1,5-AF-AGEs) [15,20,21,22,23,24,25,26,27,28,29,30], and many studies have reported the measurement of various other AGE types. Whether this novel slot blot method can measure other AGEs remains unclear. Although only the quantification of AGEs in cell lysates and tissue lysates has been reported for novel slot blot [15,20,21,22,23,24,25,26,27,28,29,30], ELISA can measure AGEs in serum [5,61,62,63,64,65,66,67], plasma [68], urine [69], saliva [5], supernatants [9,62], cell lysates [8,70], and tissue lysates [8]. It is not certain whether the AGEs in the serum, plasma, urine, and saliva can be measured using the novel slot blot. Furthermore, if researchers aim to assess AGEs generation using only fluorescence or absorbance (e.g., the inhibition example), ELISA is sufficient for this purpose.

### 4.5. Comparison with GC–MS

Although GC–MS was developed approximately 70 years ago, it is still used for the quantification of low-molecular weight compounds [10]. The target signal of the compounds and their degradation are detected as m/z, and the limitation of the range is 100–1000 Da [74]. Therefore, the molecular weights of the sample compounds must be less than 1000 Da. Furthermore, samples must be volatile when GC–MS identification is performed; the temperature conditions for detection are approximately 300–500 °C [10,74,75]. Although some types of AGEs have been detected and quantified using GC–MS, high molecular AGEs (e.g., CML-modified BSA [8], MGO-modified BSA [9], CML-modified type I collagen [71], and CML-modified HSA [71]) cannot be measured. Therefore, some researchers have performed hydrolysis of AGE-modified peptides or proteins and esterification of free AGEs (the low molecular compounds having the AGE structure) (Figure 6) [10,74,75,76]. Esterification of the compounds produces the derived compounds, which must be volatile. A common condition of hydrolysis is treatment with 6 M hydrochloride, and esterification involves treatment with ethereal diazomethane [75] and 2 M hydrochloride/methanol [10,74]. If the condition of hydrolysis and esterification of free AGEs are successful, their retention time and mass spectrum must be previously analyzed for quantification. Although researchers can analyze the molecular structure using parent and fragment peaks of the mass spectrum, which are the only data obtained from GC–MS, they are unable to identify the compounds. Therefore, AGEs quantification must be performed for compounds with retention time and mass spectrum data. Furthermore, an internal standard is needed for quantification [75,76].

In 1997, Requena et al. quantified CML and CEL in rat cell ghosts and human urine [76]. When the target AGE-modified proteins are hydrolyzed and free AGEs are collected and derivatized, urine can be sampled [76,77,78]. Internal standard AGEs are recommended for deuterium-labeled AGEs because (i) the parent and fragment ions can be distinguished from those of unlabeled AGEs (the target of AGE quantification), and (ii) the retention time of internal AGEs matches that of unlabeled AGEs [77,78]. Baskai et al. prepared three types of deuterium in CML and one type of deuterium in CEL [77]. In the GC–MS method, unlabeled CML and its deuterium, and unlabeled CEL and its deuterium were analyzed. As each parent and fragment peak of the mass spectrum were produced by deuterium compounds and their derivatives, their identification was easy. Furthermore, Baskai et al. quantified CEL in the urine of obese ZSF1 rats by analyzing unlabeled CEL-ester derivative (CEL-ester derivative-d_0_) in the urine and deuterium CEL-ester derivative (CEL-ester derivative-d_6_) (Figure 7) [78].

The novel slot blot is able to measure AGE-modified proteins collected from cells or tissues and does not require pretreatments, such as hydrolysis and ester derivatization. In addition, if the researcher cannot identify the structure of the target AGEs, anti-AGE antibodies can be used to recognize the epitopes of AGEs. Therefore, researchers can perform the novel slot blot even without any data on the mass of glycation (e.g., glucose, fructose, and their intermediates). Based on these characteristics, the novel slot blot is better than GC–MS. GC–MS only targets free AGEs, such as the free type of CML, and quantification of AGE-modified proteins cannot be performed. Furthermore, only AGEs whose structures are known and recorded in a database can be targeted (if researchers can analyze structures with GC–MS, they must provide validity using other methods, such as nuclear magnetic resonance (NMR)). However, GC–MS can provide data on the mass of regular and degraded AGEs because it can detect their ion peaks. On the contrary, the novel slot blot analysis cannot provide any data on the mass and structure of AGEs. GC–MS can perform absolute quantification when an internal standard is provided. Quantification using the novel slot blot was conducted based on the luminance (intensity) of standard AGEs (e.g., TAGE-BSA and 1,5-AF-AGE-BSA). If specific AGE proteins such as anti-CML-modified actin antibodies are present, the slot blot may be able to measure it. However, currently available anti-AGEs antibodies can recognize each AGE structure and the various proteins that possess those structures. Therefore, the slot blot can quantify crude AGE-modified proteins but not specific AGE-modified proteins. For this purpose, GC–MS is more suitable than slot blotting.

### 4.6. Comparison with MALDI–MS and LC–ESI–MS

MALDI–MS [17,18,44,45,46] and ESI–MS [19,43,48,49,50] have progressed and developed proteomic investigations in this century. In both MALDI–MS and ESI–MS, the peptides in samples digested with enzymes (the major enzyme being trypsin) can be identified and detected. Because enormous amounts of amino acid sequence data have been accumulated in a database, and various software packages have been able to calculate the peptide weights, MS/MS analysis has successfully and accurately provided the parent signals of various peptides and fragments. However, the limitations faced in detecting the molecular weight of peptides are greater for MALDI–MS than for ESI–MS [11,49,50]. Further, ESI–MS is appropriate for analyzing crude peptides as it can be combined with LC, and the molecular weight data and retention time of peptides can be evaluated [48,49,50]. Recently, LC–MALDI–MS was developed and used [79,80], although it has been used less frequently than LC–ESI–MS.

Nevertheless, both MALDI–MS and LC–ESI–MS have similar limitations. If the data of the peptide sequence (containing modified peptides, such as by acetylation, methylation, glycosylation, and glycation) were not accumulated in the database, the software is unable to identify them. Therefore, the proteins for which the peptides are provided to the MS equipment remain unknown. This feature may be problematic in the identification and quantification of AGEs. As insufficient data for free AGEs (e.g., CML, CEL) and AGE-modified peptides are registered in the MS analysis software, many AGE-modified proteins cannot be identified (Figure 8) or quantified. However, some researchers have identified AGE-modified proteins based on the data analyzed for each peak of peptide ions and modified peptides. In this examination, information on the fragment ion peak and NMR of known AGEs was considered crucial [81,82,83]. Novel AGEs have been produced from lactaldehyde using LC–ESI–MS [84]. Lactose and BSA were dissolved in PBS and then incubated, following which lactaldehyde-derived BSA was prepared. To collect lactaldehyde-derived compounds, an enzymatic treatment was performed. Based on the ion peak data of LC–ESI–MS and NMR, the identification of novel AGEs was successfully performed. In the future, these data may be accumulated with software packages used for proteomic analysis.

The investigation with MALDI–MS: Perween et al. isolated fibrinogen from plasma and incubated it with MGO for 7 d to prepare MGO-fibrinogen [85]. Further, they identified the site of MGO modification. Ghosh et al. analyzed HSA under diabetic conditions and identified the site of modification as well [86]. Zhang et al. revealed the modification of AGEs (1,4-alkyl-2-formyl-3, 4-glycosyl-pyrrole (AFGP), imidazolone B, pyrraline, CML, crossline, and pentosidine) in human β-2-microglobulin, rat type II randorine receptor calcium-release channel (RyR2), and sarco(endo)plasmic reticulum (SERCA2a) [87]. Iles et al. also revealed that glycated HSA levels were higher in patients with COVID-19 [11]. Human β-2-microglobulin was isolated from humans and subjected to high-glucose conditions for 60 d. Rat RyR2 and SERCA2a were isolated from streptozotocin-induced diabetic rats and age-matched controls. They revealed one of the mass spectra forms of β-2-microglobulin in control and high glucose conditions, and rat RyR2 and SERCA2a in the control and diabetic models; the peak of AGE-modified peptides was clearly detected against the control. The ion peak intensity indicated the tendency of each AGE-modified protein to increase in the high-glucose condition or diabetic model. However, quantification using the ion peak is not recommended because the peak intensity is not necessarily directly proportional.

The investigation with LC–ESI–MS: LC–ESI–MS can quantify compounds based on the peak of the retention time. In previous proteomic analyses with LC–ESI–MS, deuterium-labeled proteins were used as internal standards [88,89]. Unlabeled and deuterium-labeled peptides, prepared with enzymatic digestion (e.g., trypsin digestion), were detected and identified, and their retention times were matched. This method can be applied to AGEs quantification. However, AGE-modified peptides cannot be identified using the general software for ESI–MS and MALDI–MS (Figure 8). Therefore, this quantification method was modified for the previous GC–MS and ESI–MS. While the amount of AGE-modified proteins could not be quantified, that of free AGEs (e.g., free CML) in the samples could be.

Kuhang et al. quantified CML, which was removed from proteins with trichloroacetic acid, in the plasma of patients with diabetes and uremia [90]. Based on their investigations, Mastrocola et al. quantified free CML (prepared from hydrolysis with 6 M HCl of CML-modified proteins) in murine serum and skeletal muscle (Figure 9) [91,92]. The units were ng/mL for CML in plasma and μg/g of tissue for CML in skeletal muscle. Because the mass of only free CML was measured, that of CML-modified proteins may have been greater than the acquired data. Quantification of CML is beneficial for research on lifestyle-related diseases, such as diabetes and sarcopenia [90,91,92]. Studies show the use of a method in which the value of quantified free AGEs (e.g., CML, CMA, and MG-H1) prepared via hydrolysis of AGE-modified proteins with HCl was normalized with the lysine content [12,71,73,93,94,95]. Kinoshita et al. prepared CMA-modified and CML-modified type I collagen and HSA, and then performed hydrolysis to collect free CMA and CML [71]. They used deuterium-labeled lysine (Lys) ([^13^C_6_] lysine) and arginine (Arg) ([^13^C_6_] arginine) as the internal controls. Further, they quantified the CMA and CML (pmol/30 μg protein). However, they also calculated another value of CMA and CML (pmol/mol Lys and pmol/mol Arg). If the number of lysine residues in each protein had been normalized, they could have been determined in the protein database. Because deuterium-labeled lysine was used as an internal control, the results identified and quantified for lysine in the experiment with LC–ESI–MS were required. As one of the peptides, but not all peptides in proteins, can be detected with LC–ESI–MS equipment and software, the quantification of lysine in the detected peptides may be required.

Furthermore, CML, CEL, MG-H1, and CMA were quantified in murine skin (mmol/μg mouse skin) [71]. Although the calculated values were in the units of mmol/μg mouse skin, they also calculated them in the units mmol/mol Lys or mmol/mol Arg. This method of quantification revealed the tendency of the generation and accumulation of AGEs in murine skin. The results demonstrate the importance of using an appropriate standard for normalization in quantification experiments. Ban et al. prepared CML- and MG-H1-modified BSA, quantified CML (mmol/mol Lys) and MG-H1 (mmol/mol Arg) [73], and found that natural products inhibited the generation of MG-H1 from ribose and BSA. Ohono et al. quantified CML in the eye lenses of diabetic model rats [12]. Interestingly, they proved that only glucose–lysine could be identified and quantified in the samples although fructose–lysine existed in the same samples. As the molecular weights and retention times of glucose–lysine and fructose–lysine are similar, researchers considered that fructose–lysine hindered the quantification of glucose–lysine. However, Ohono et al. also found that fructose–lysine was completely transferred to fructosin with HCl hydrolysis, whereas glycose–lysine was resistant to HCl hydrolysis [12]. Glucose–lysine is a fructose-derived AGE that can be used to quantify glucose–lysine and fructosin in the eye lenses and kidneys of diabetic model rats.

In another study, Suzuki et al. treated the murine osteoblastic cell line MC3T3-E1 with glycol-aldehyde and quantified CML (mmol/mol Lys) [93]. Kato et al. quantified CML, CEL, MG-H1, and CMA (mmol/mol Lys or mmol/mol Arg) in human serum (healthy controls, and patients with nephropathy or diabetic nephropathy) [94]. Katsuta et al. quantified CML and MG-H1 (mmol/mol Lys) in the brain, liver, and kidneys of 4-, 11-, and 96-week-old C57BL/6 J mice [95]. They showed that senescence induced the upregulation of CML in the brain and CML and MG-H1 in the kidney. If rats gain excess glucose or fructose, senescence may induce AGEs accumulation in their brain and kidneys.

Although the identification of AGE-modified peptides has been difficult, Papadaki et al. successfully identified 23 MGO-modified peptides in non-failing controls and 32 MGO-modified peptides in patients with dbHF [16]. Furthermore, nine MGO-modified peptides were identified only in dbHF. A graph of total MS/MS spectral counts was drawn. As the quantification of MGO-modified peptides was successful, this analysis should include the quantification of AGEs. Furthermore, Senavirathna et al. treated ^13^C isotopic glyceraldehyde against PANC-1 and HPDE (human normal pancreatic epithelial cells), similar to our investigation [15,96], and quantified glyceraldehyde-derived pyridinium compound (GLAP), argpyrimidine, and MG-H1. First, they used Trans-Proteomic Pipeline software (TPP, version v5.1.0), which could search lysine AGE modification-GLAP (variable), arginine AGE modification-argpyrimidine (variable), and arginine AGE modification-MG-H1 (variable). Therefore, they could identify these AGE-modified proteins and detect them using LC–ESI–MS. Second, they drew a graph of the number of GLAP, argpyrimidine, and MG-H1 in PANC-1 and HPDE, and revealed that the ratio of accumulation of these GA-AGEs was affected by the concentration of glyceraldehyde (0, 1, 2, and 4 mM). Third, this provided the scope to develop an appropriate technology to identify AGE-modified peptides. Fourth, GLAP was generated from glyceraldehyde [97,98,99]. Although argpyrimidine and MG-H1 were produced from glyceraldehyde in tubes [100,101], there are many reports that argpyrimidine [102,103,104,105] and MG-H1 [105,106,107,108] generate MGO; hence, they are generally categorized as MGO-AGEs [109]. However, this study revealed that argpyrimidine and MG-H1 were generated from glyceraldehyde in PANC-1 and HPDE cells. Finally, other structures of GA-AGEs (unidentified in the cell) [1,110,111] could be included in the modification data, identified, and quantified.

The characteristics of LC–MALDI–MS and LC–ESI–MS are similar to those of GC–MS. Therefore, the comparison of advantages between these methods and our slot blot are similar as well. Whereas the novel slot blot could quantify whole crude AGEs (AGE-modified proteins) using standard AGEs, this was not possible for the other methods. Furthermore, slot blotting does not require any pretreatment, such as enzymatic digestion. Our slot blot analysis also does not require the data of the mass and structure of glycation. These characteristics of the slot blot make it superior to the other techniques. However, the above methods can perform absolute quantification with standard AGEs, such as CML, CEL, MG-H1, and CMA. They can provide information on the mass of regular and degraded AGEs as well as GC–MS. This is an advantage that the novel slot blot does not possess. Furthermore, there are some reports that AGE-modified peptides were identified and quantified using other methods [16,95]. In the future, LC–MALDI–MS and LC–ESI–MS may completely identify all AGE-modified peptides in the sample. If internal AGE-modified peptides are provided, then each of them can be used for absolute quantification.

## 5. Conclusions

The novel slot blot assay has some advantages and disadvantages compared to other methods. Although the slot blot technology is not essential for AGEs investigation, it can provide certain information that other methods cannot. It is most beneficial for researchers to simultaneously perform the novel slot blot and other methods for AGEs quantification. As researchers should select the most appropriate method to achieve their experimental objectives, the novel slot blot can be a valuable addition to the available methods. Therefore, it can serve as a useful technique for the quantification of intracellular AGEs.

## Figures and Tables

**Figure 1 metabolites-13-00564-f001:**
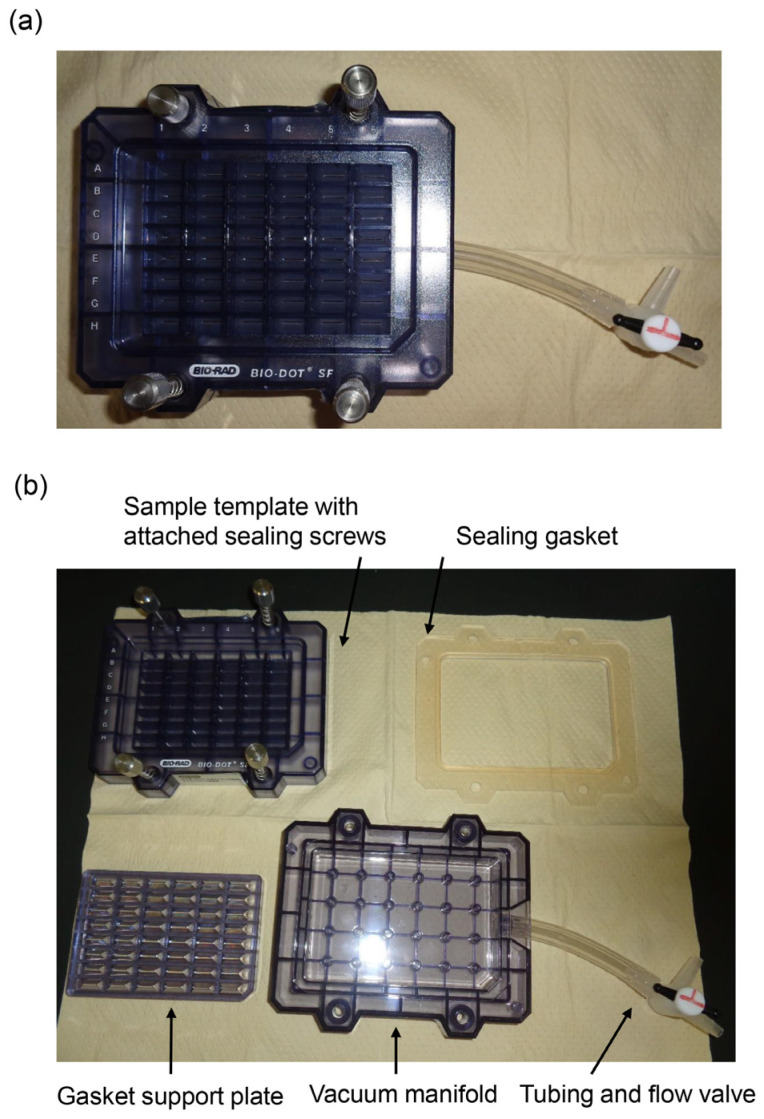
Bio-Dot SF Microfiltration Apparatus (slot blot apparatus with 48 lanes). (**a**) Assembly of the apparatus. (**b**) Parts of the disassembled apparatus.

**Figure 2 metabolites-13-00564-f002:**
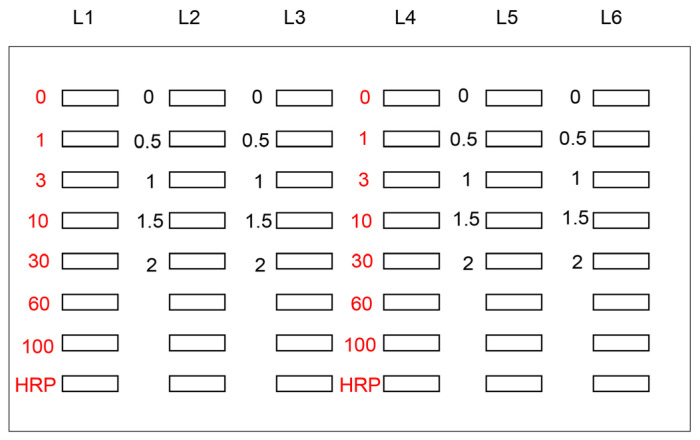
Setup used for the application of standard TAGE-BSA, HRP marker solution, and sample solution onto the PVDF membrane [27]. White open squares indicate the slot lane. L1, L4: 0, 1, 3, 10, 30, 60, and 100 ng of TAGE-BSA aliquots and HRP marker solution. L2, L3, L5, L6: cell lysate samples of C2C12 cells treated with 0, 0.5, 1, 1.5, and 2 mM glyceraldehyde for 24 h.

**Figure 3 metabolites-13-00564-f003:**
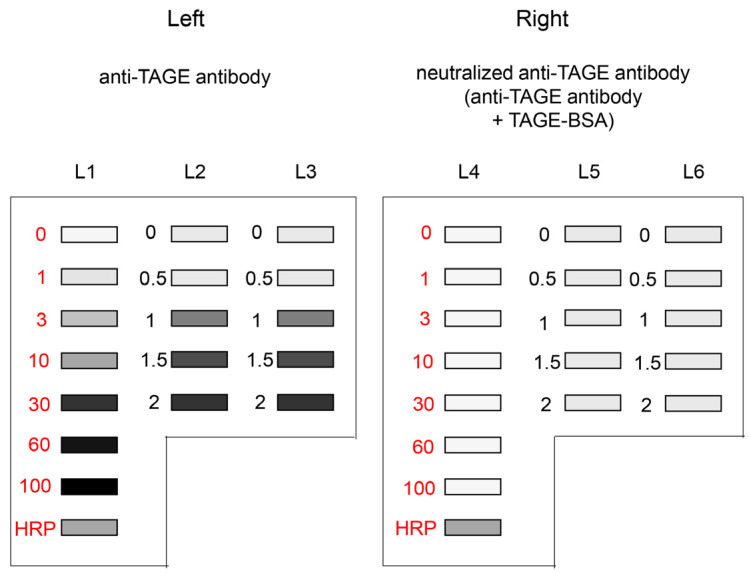
Chemiluminescence detection of standard TAGE-BSA aliquots, HRP marker solution, and sample solution on a PVDF membrane [27]. Both membrane sides were simultaneously exposed using Fusion FX. (**Left**) (L1–L3): Anti-TAGE antibody was probed onto the PVDF membrane. (**Right**) (L4–L6): Neutralized anti-TAGE antibody was probed onto the PVDF membrane. Closed gray and black squares indicate the bands on the PVDF membrane. L1, L4: 0, 1, 3, 10, 30, 60, and 100 ng of TAGE-BSA aliquots and HRP marker solution. L2, L3, L5, L6: cell lysate samples of C2C12 cells treated with 0, 0.5, 1, 1.5, and 2 mM glyceraldehyde for 24 h.

**Figure 4 metabolites-13-00564-f004:**
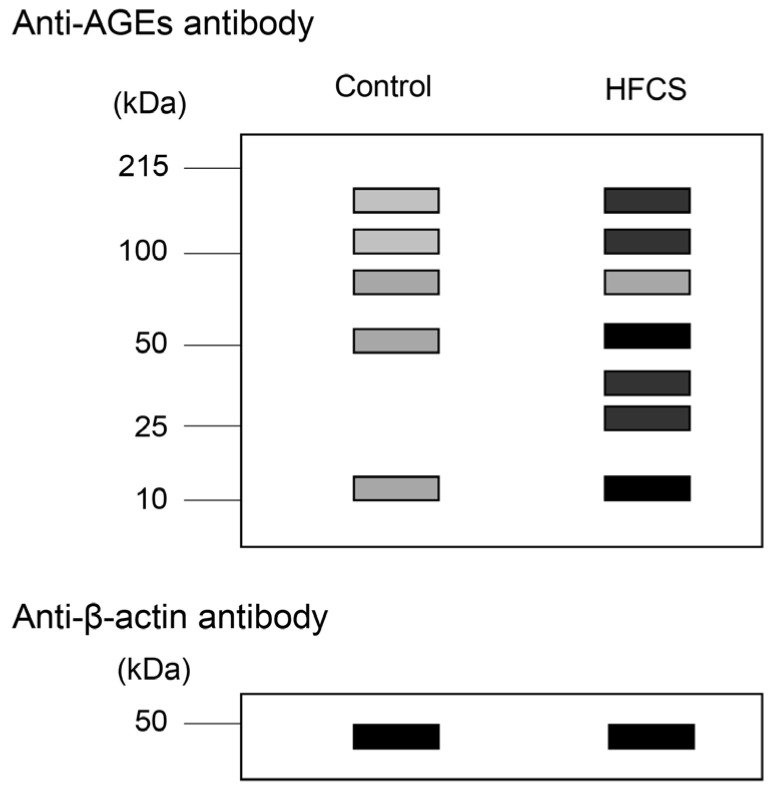
Model image of western blot with anti-AGE antibody and anti-β-actin. Equal amounts of lysates of hepatic tissues were used. Control: hepatic tissues of rats that had drunk water. HFCS: hepatic tissues of rats that had drunk high-fructose corn syrup (HFCS). Closed gray and black squares indicate the bands on the membrane. The luminance of each band was analyzed and normalized with β-actin. Finally, luminance amounts were quantified.

**Figure 5 metabolites-13-00564-f005:**
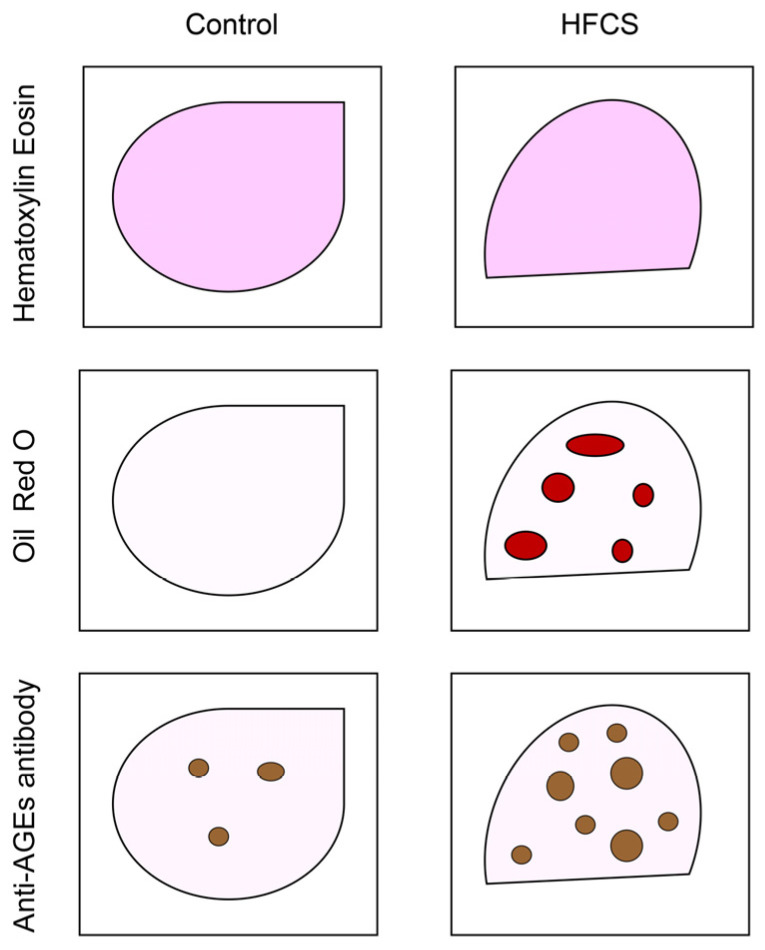
Model image of the immunostaining of rat hepatic tissue with hematoxylin and eosin, oil red O, and anti-AGE antibody. Control: hepatic tissues of rats that had drunk water. HFCS: hepatic tissues of rats that had drunk high-fructose corn syrup (HFCS). Closed red circles are steatosis drops. Closed brown circles are the areas where anti-AGE antibody was probed.

**Figure 6 metabolites-13-00564-f006:**
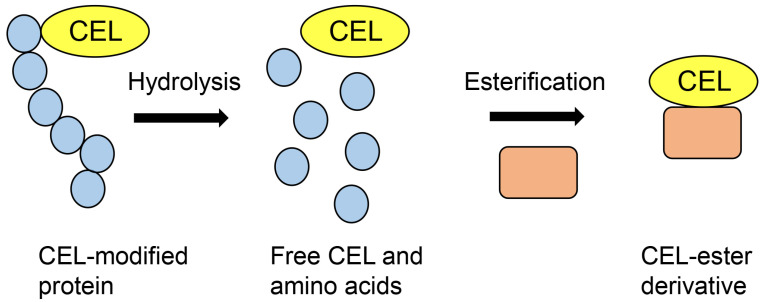
Model image of preparation of CEL-ester derivative for GC–MS analysis. Closed blue circles are amino acids. Closed peach squares are the compounds which have hydroxyl groups.

**Figure 7 metabolites-13-00564-f007:**
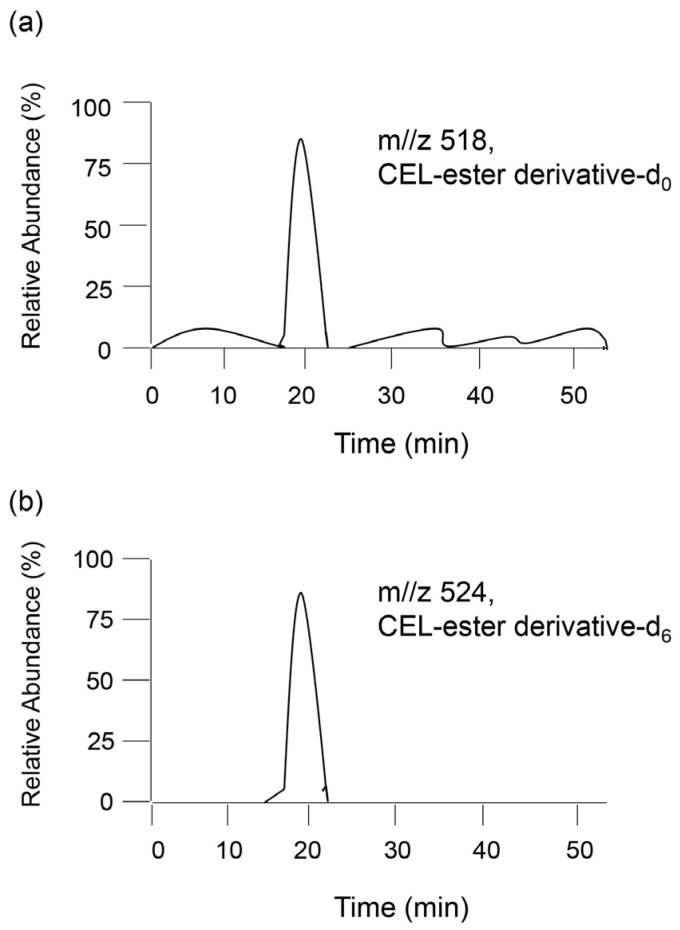
Model image of detection of (**a**) unlabeled CEL-ester derivative (CEL-ester derivative-d_0_) in a sample and (**b**) deuterium CEL-ester derivative (CEL-ester derivative-d_6_) as the standard for GC–MS analysis.

**Figure 8 metabolites-13-00564-f008:**
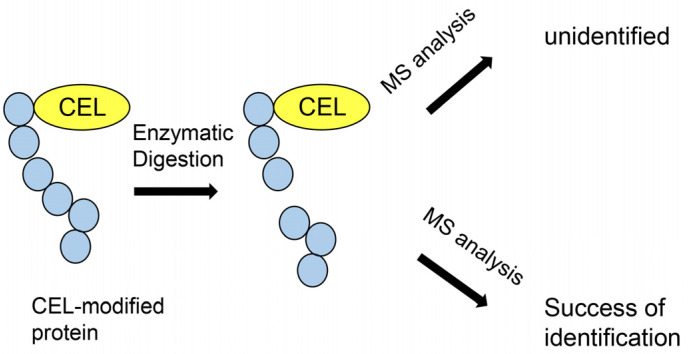
Model image of MALDI–MS or ESI–MS analysis of CEL-modified protein.

**Figure 9 metabolites-13-00564-f009:**
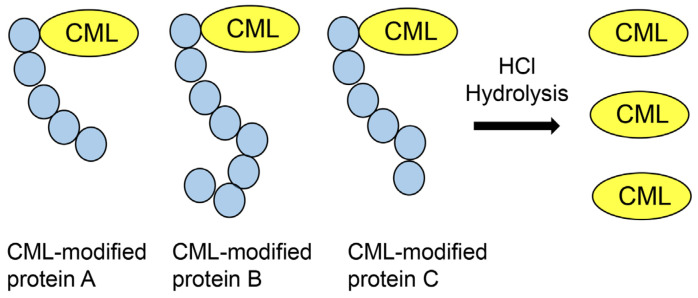
Hydrochloride (HCl) hydrolysis of various CML-modified proteins and collection of free CML.

**Table 1 metabolites-13-00564-t001:** Solutions used for the preparation of lysis buffer [15,20,21,22,23,24,25,26,27,28,29,30].

Solution A	Solution B	Solution C	Solution D
30 mM Tris	1 protease inhibitor cocktail tablet/2 mL	27 mM Tris	30 mM Tris
7 M urea		6.3 M urea	7 M urea
2 M thiourea		1.8 M thiourea	2 M thiourea
4% CHAPS		3.6% CHAPS	4% CHAPS
		10% Solution B	4% Solution B
(pH 8.5)		(pH 8.5)	(pH 8.5)

**Table 2 metabolites-13-00564-t002:** References used for preparing Solution C and D.

Solution	References
Solution C	[15,22,23,24,25,26,27,29]
Solution D	[20,21,28,30]

**Table 3 metabolites-13-00564-t003:** References for the cell types lysed with Solution C and D.

Cell Type	References
Tissue	[23,29]
Primary cells	[23,24]
Normal cells	[25]
Differentiated from iPS cells	[22]
Cell line	[15,20,21,26,27,28,30]

iPS cells, induced pluripotent stem cells.

**Table 4 metabolites-13-00564-t004:** References for the organ sources of cells lysed with Solution C and D.

Organ	References
Liver	[20,21,22,23,29,30]
Heart	[24,25]
Pancreas	[15,26]
Skeletal muscle	[27]
Bone	[28]

**Table 5 metabolites-13-00564-t005:** Competition between cell viability and intracellular TAGE.

Cell Type	Treatment	Cell Viability (%)	TAGE (μg/mg Protein)	Reference
Rat primary cardiomyocytes	0 mM, 24 h	100	0	[24]
	2 mM, 24 h	13	28.7	
	4 mM, 24 h	0	38.5	
	4 mM, 0 h	100	0	
	4 mM, 6 h	39	12.0	
	4 mM, 12 h	7	34.2	
Human normal cardiac fibroblasts	0 mM, 24 h	100	0.3	[25]
	1 mM, 24 h	51	2.8	
	1.5 mM, 24 h	34	4.0	
	2 mM, 24 h	14	8.1	
hiPSC-HLC	0 mM, 24 h	100	0.6	[22]
	4 mM, 24 h	20	5.4	
PANC-1	0 mM, 24 h	100	0	[15]
	2 mM, 24 h	40	6.4	
	4 mM, 24 h	6	21.2	
1.4E7	0 mM, 24 h	100	0.2	[26]
	1.5 mM, 24 h	61	3.0	
	2 mM, 24 h	53	5.2	
	2.5 mM, 24 h	33	6.3	
	3 mM, 24 h	12	8.8	
C2C12	0 mM, 24 h	100	0	[27]
	1.5 mM, 24 h	48	6.0	
	2 mM, 24 h	5	15.9	
Hep3B	0 mM, 24 h	100	0	[54]
	2 mM, 24 h	46	0.2	
	4 mM, 0 h	100	0	
	4 mM, 6 h	65	0.2	
	4 mM, 12 h	11	0.4	
	4 mM, 24 h	8	0.9	

The cells were treated with glyceraldehyde in vitro. Cell viability was assessed using the WST-8 method [15,22,24,25,27], Cell Titer-Gro method [26], and WST-1 method [54]. Intracellular TAGE were quantified using TAGE-BSA with the novel slot blot [15,22,24,25,26,27] and Takino et al.’s slot blot [54]. The value was statistically analyzed, showing a significant increase compared to that in the control (0 mM or 0 h) [15,22,24,25,26,27]. The unit used was μg per mg protein. In Takino et al.’s study, the results were expressed as arbitrary units (U): 1 U corresponded to 1 μg of TAGE-BSA, and 1 mU corresponded to 1 ng of TAGE-BSA [54]. They expressed the value (mU) in the graph of intracellular TAGE per 30 μg of protein, because they applied 30 μg of protein to the PVDF membrane [54]. Therefore, I converted the values of their data to a single unit, that is, μg/mg protein.

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
