# Peer review of "Is the Novel Slot Blot a Useful Method for Quantification of Intracellular Advanced Glycation End-Products?"

_metabolites, 2023, doi:10.3390/metabo13040564_

Round 1
Reviewer 1 Report
Congratulation to the author for the tremendous work and comprehensive review and comparison about detection methods for AGE.
I have one suggestion may be considered. Maybe the author could consider discussing about glycan mass and novel slot blot method.
Author Response
Response Letter to Reviewers’ Comments
Dear Reviewer 1,
Thank you for giving me the opportunity to submit a revised draft of my manuscript titled “Is the Novel Slot Blot a Useful Method for Quantification of Intracellular Advanced Glycation End-Products?” to the journal Metabolites (manuscript ID: 2306811). I appreciate the time and effort that the reviewers have dedicated to providing their valuable feedback on my manuscript. I am grateful to the reviewers for their thoughtful suggestions and insightful comments on my paper, which have enriched the manuscript and produced a better and more balanced account of the research.
I have incorporated changes in the manuscript to reflect the suggestions provided by the reviewers. For the reviewers’ ease, I used yellow highlights to indicate the revised sentences in the manuscript. I also changed the font color of my responses below to red.
Please note that there were two numerical errors in Table 5 “Competition between cell viability and intracellular TAGE” in the first row. These errors have been rectified and highlighted in yellow color. I also inserted a new Ref. 2 and renumbered the previous Ref. 2 to 109, Ref. 109 to 110, and Ref. 110 to 111.
Here is a point-by-point response to the reviewers’ comments and concerns.
Comments of Reviewer 1:
Congratulation to the author for the tremendous work and comprehensive review and comparison about detection methods for AGE.
Comment 1: I have one suggestion may be considered. Maybe the author could consider discussing about glycan mass and novel slot blot method.
Response: Following your suggestions, I have added the following sentences in Section 4.5. “Comparison with GC-MS section”:
“Therefore, researchers can perform the novel slot blot even without any data on the mass of glycation (e.g. glucose, fructose, and their intermediates).” (Lines 514–516)
“However, GC-MS can provide data on the mass of regular and degraded AGEs because it can detect their ion peaks. On the contrary, the novel slot blot analysis cannot provide any data on the mass and structure of AGEs.” (Lines 521–523)
I have also discussed these points in relation to MALDI-MS and ESI-MS in Section 4.6 “Comparison with MALDI-MS and LC-ESI-MS”, by adding the following sentences:
“Our slot blot analysis also does not require the data of the mass and structure of glycation.” (Lines 682-683)
“They can provide information on the mass of regular and degraded AGEs as well as GC-MS. This is an advantage that the novel slot blot does not possess.” (Lines 686–687)

Reviewer 2 Report
Reviewing the manuscript entitled, “Is the Novel Slot Blot a Useful Method for Quantification of Intracellular Advanced Glycation End-Products? by Takata T et al., this manuscript is a review focusing on Blot technology of TGEs. Although this is interesting research, it seems that s content does not match the aim of the journal.
In the case of a review, author should clearly describe the historical background of AGE quantification and the significance and importance of AGE quantification.
Additionally, authors should describe the importance of Blot technology in AGE research.
This version is all about making samples of AGE-blot and describing the method.
The importance of this review manuscript is understandable if BLOT technology is essential for AGE research. Unfortunately, that is not confirmed in this version.
This looks fine for reviews on simple Blot techniques, not AGE studies.
Also, the conclusions do not appear to be particularly in need of reviewing manuscript.
Author Response
Response Letter to Reviewers’ Comments
Dear Reviewer 2,
Thank you for giving me the opportunity to submit a revised draft of my manuscript titled “Is the Novel Slot Blot a Useful Method for Quantification of Intracellular Advanced Glycation End-Products?” to the journal Metabolites (manuscript ID: 2306811). I appreciate the time and effort that the reviewers have dedicated to providing their valuable feedback on my manuscript. I am grateful to the reviewers for their thoughtful suggestions and insightful comments on my paper, which have enriched the manuscript and produced a better and more balanced account of the research.
I have incorporated changes in the manuscript to reflect the suggestions provided by the reviewers. For the reviewers’ ease, I used yellow highlights to indicate the revised sentences in the manuscript. I also changed the font color of my responses below to red.
Please note that there were two numerical errors in Table 5 “Competition between cell viability and intracellular TAGE” in the first row. These errors have been rectified and highlighted in yellow color. I also inserted a new Ref. 2 and renumbered the previous Ref. 2 to 109, Ref. 109 to 110, and Ref. 110 to 111.
Here is a point-by-point response to the reviewers’ comments and concerns.
Comments of Reviewer 2:
Comments and Suggestions for Authors
Reviewing the manuscript entitled, “Is the Novel Slot Blot a Useful Method for Quantification of Intracellular Advanced Glycation End-Products? by Takata T et al., this manuscript is a review focusing on Blot technology of TGEs. Although this is interesting research, it seems that s content does not match the aim of the journal.
Comment 1: In the case of a review, author should clearly describe the historical background of AGE quantification and the significance and importance of AGE quantification.
Response: I agree with the comment by Reviewer 2 and, therefore, rewrote some sentences in the Introduction section. I explained the following: (Lines 32–41)
- Researchers have measured various AGEs in the blood and analyzed their relation-ships with many diseases, particularly lifestyle-related diseases (e.g. diabetes, cardio-vascular disease).
- In fact, some AGEs in the blood may serve as biomarkers for some diseases.
- As dietary AGEs are taken into the body and are known to cause health effects through receptors for AGEs, they have been extensively studied and quantified by researchers.
- On the contrary, the quantification of intracellular AGEs in some organs during clinical investigations has proven challenging. However, their quantification in vitro or in animal models has been conducted to investigate their impacts on the regulation of certain components (e.g. proteins and reactive oxygen species) in diseased states (e.g. hyperglycemia and diabetes).
Comment 2: Additionally, authors should describe the importance of Blot technology in AGE research.
Response: Following your suggestion, I added these sentences in the Introduction section:
“The slot blot analysis is a useful technology because it can detect AGEs through the use of anti-AGE antibodies, which can be easily obtained or prepared. Furthermore, the analytical samples do not require any pretreatment, such as acid hydrolysis, derivatization, or enzymatic digestion. However, the structure of individual AGEs cannot be identified through this technique.” (Lines 49–53)
Comment 3: This version is all about making samples of AGE-blot and describing the method. The importance of this review manuscript is understandable if BLOT technology is essential for AGE research. Unfortunately, that is not confirmed in this version.
Response: Thank you for this pertinent comment. I have mentioned in the manuscript that the slot blot technology is not essential for AGE research. However, it is indeed beneficial for researchers to simultaneously perform slot blot analysis and other methods for AGE quantification. I compared the methodology, strengths, and weaknesses of the novel slot blot with other methods, and I concluded (in the Conclusion section) that the novel slot blot has certain advantages over other methods, such as providing unique information on the AGEs in the sample that cannot be obtained using other methods (Lines 694–697).
For example, if a researcher finds that an individual AGE-modified protein (e.g. argpyrimidine-modified heat shock protein 90) has increased in the organs in diabetes model rats, it is possible that the total argpyrimidine-modified proteins may have been decreased. Therefore, if the researcher employs the simultaneous use of multiple analytical techniques, they can obtain more valuable data in this experiment, rather than only focusing on the relationship between argpyrimidine-modified heat shock protein 90 and diabetes.
Comment 4: This looks fine for reviews on simple Blot techniques, not AGE studies.
Response: Thank you for this pertinent comment. I have considered the fact that the novel slot blot can measure other proteins as well (e.g. phosphorylated protein, methylated protein, acetylated protein, and glycosylated protein). However, 12 studies reporting the quantification of AGEs with the novel slot blot have been published from 2017–2022, and there is no report of other type of proteins being analyzed with it. Therefore, I only discussed the measurement of AGEs with the novel slot blot.
Comment 5: Also, the conclusions do not appear to be particularly in need of reviewing manuscript.
Response: I have revised the Conclusion section to make it more relevant to the review. The revised Conclusion is as follows:
“The novel slot blot assay has some advantages and disadvantages compared to those of other methods. Although the slot blot technology is not essential for AGE investigation, it can provide certain information that other methods cannot. It is most beneficial for researchers to simultaneously perform the novel slot blot and other methods for AGE quantification. As researchers should select the most appropriate method to achieve their experimental objectives, we consider the novel slot blot to be a valuable addition to the available methods. Therefore, it can serve as a useful technique for the quantification of intracellular AGEs.” (Lines 693–700)

Reviewer 3 Report
I have minor issues with the paper.
1. Line 9: The article has been written by a single author, so the phrase "We have reported... "needs paraphrasing. This also applies to "we hypothesized.." in line 179, and “we did not consider..” in line 335.
2. Line 10: should be: (the abbreviation…)
3. Line 86: “Although Ghiani et al. semi-quantified allergens in the samples,…” - the verb is missing here.
4. Lines 101-102: Bio-Dot SF Microfiltration Appa- 101 ratus (Cat. no.: 170-6452) - the name of the manufacturer is missing here, it is Bio-Rad.
5. Section 3.2.: If the compositions and concentrations of solutions A, B, C and D are given in Table 1, then the description of their preparation in this paragraph is incorrect. In my opinion, instead of: "First, 30 mM Tris, 7 M urea, 2 M thiourea, and 4% CHAPS were dissolved in ultra-pure water (Solution A, Table 1)." should read: Tris, urea, thiourea and CHAPS were dissolved in ultra-pure water to make 30 mM Tris, 7 M urea, 2 M thiourea, and 4% CHAPS (Solution A, Table 1). Similarly in other cases. It would also be worth adding where the individual reagents were purchased, especially when it comes to a protease inhibitor cocktail.
6. Line 129: should be: Solution C…
7. Lines 129-130: “for novel slot blots of 8 and 4 129 (Table 2)”- I don't understand what this term means.
8. Line 208: Please enter the name(s) of the manufacturer(s) of the chemiluminescent kit and anti-TAGE antibody.
9. Lines 228-229: Has only one component (A right - B right) been multiplied by the calibration corrected value? Or did it apply to both components?
10. Table 5: rat primary cardiomyocytes - Why does 4 mM, 24 h treatment appear twice in the list? In addition, the TAGE values given for them are different, although the data come from one reference [24].
11. Table 5: Hep3B - Information on 4 mM, 24 h treatment should be placed at the end of the list.
12. Figure 7 legend: Last two sentences to be deleted.
13. Lines 602-604: I suggest you use the term "glucose" instead of "glycose". Especially since this term appears more frequently in the manuscript.
Author Response
Response Letter to Reviewers’ Comments
Dear Reviewer 3,
Thank you for giving me the opportunity to submit a revised draft of my manuscript titled “Is the Novel Slot Blot a Useful Method for Quantification of Intracellular Advanced Glycation End-Products?” to the journal Metabolites (manuscript ID: 2306811). I appreciate the time and effort that the reviewers have dedicated to providing their valuable feedback on my manuscript. I am grateful to the reviewers for their thoughtful suggestions and insightful comments on my paper, which have enriched the manuscript and produced a better and more balanced account of the research.
I have incorporated changes in the manuscript to reflect the suggestions provided by the reviewers. For the reviewers’ ease, I used yellow highlights to indicate the revised sentences in the manuscript. I also changed the font color of my responses below to red.
Please note that there were two numerical errors in Table 5 “Competition between cell viability and intracellular TAGE” in the first row. These errors have been rectified and highlighted in yellow color. I also inserted a new Ref. 2 and renumbered the previous Ref. 2 to 109, Ref. 109 to 110, and Ref. 110 to 111.
Here is a point-by-point response to the reviewers’ comments and concerns.
Comments of Reviewer 3:
Comments and Suggestions for Authors.
I have minor issues with the paper.
Comment 1: Line 9: The article has been written by a single author, so the phrase "We have reported... "needs paraphrasing. This also applies to "we hypothesized.." in line 179, and “we did not consider..” in line 335.
Response: According to your suggestion, I have rephrased all the relevant instances in the manuscript to show that the review was written by a single author.
Comment 2: Line 10: should be: (the abbreviation…)
Response: I have revised it to simply say “(TAGE)” (Line 10).
Comment 3: Line 86: “Although Ghiani et al. semi-quantified allergens in the samples,…” - the verb is missing here.
Response: I have rewritten this sentence to “Although Ghiani et al. performed semi-quantification of allergens” (Lines 100–101).
Comment 4: Lines 101-102: Bio-Dot SF Microfiltration Appa- 101 ratus (Cat. no.: 170-6452) - the name of the manufacturer is missing here, it is Bio-Rad.
Response: Thank you for this pertinent suggestion. I have included the location details as follows: “Bio-Rad Laboratories Inc., Hercules, CA, USA” (Line 117).
Comment 5: Section 3.2.: If the compositions and concentrations of solutions A, B, C and D are given in Table 1, then the description of their preparation in this paragraph is incorrect. In my opinion, instead of: "First, 30 mM Tris, 7 M urea, 2 M thiourea, and 4% CHAPS were dissolved in ultra-pure water (Solution A, Table 1)." should read: Tris, urea, thiourea and CHAPS were dissolved in ultra-pure water to make 30 mM Tris, 7 M urea, 2 M thiourea, and 4% CHAPS (Solution A, Table 1). Similarly in other cases. It would also be worth adding where the individual reagents were purchased, especially when it comes to a protease inhibitor cocktail.
Response: I apologize for the lack of clarity and thank the reviewer for highlighting the ambiguity in the phrasing I have used. I have revised the sentences describing the preparation of the solutions as follows:
“The components of the lysis buffer were modified with a buffer prepared for 2DE-based proteomic analyses. First, Tris (Cat. no.: 011-20095; Fujifilm Wako Pure Chemical, Osaka, Japan), urea (Cat. no.: 217-01215; Fujifilm Wako Pure Chemical), thiourea (Cat. no.: 201-17355; Fujifilm Wako Pure Chemical) and CHAPS (Cat. no.: 347-04723; DOJINDO Laboratories, Kumamoto, Japan) were dissolved in ultra-pure water to make a solution of 30 mM Tris, 7 M urea, 2 M thiourea, and 4% CHAPS (Solution A, Table 1). Second, a protease inhibitor cocktail tablet (Complete Tablets EDTA-Free, EASY pack, Cat. no.: 04-693-132-001; Roche, Penzberg, Bavaria, Germany) was dissolved in ultra-pure water (final volume: 2 mL, Solution B, Table 1 [15,24]). Finally, Solution A and Solution B were mixed together (9:1) (Solution C, Table 1). The components of Solution C were 27 mM Tris, 6.3 M urea, 1.8 M thiourea, and 3.6% CHAPS, making up the lysis buffer used for our novel slot blot analysis [15,22-27,29]. The final components of Solution D were based on the preparation of Solution C. When Tris, urea, thiourea, and CHAPS were dissolved in ultra-pure water, Solution B was added to it (Table 1) [20,21,28,30].” (Lines 123–136)
Comment 6: Line 129: should be: Solution C…
Response: I have corrected this sentence.
Comment 7: Lines 129-130: “for novel slot blots of 8 and 4 129 (Table 2)”- I don't understand what this term means.
Response: I apologize for the ambiguity in this sentence. I simply wanted to mention that the number of references used for preparing Solutions C and D were 8 and 4, respectively (Lines 150–151).
Comment 8: Line 208: Please enter the name(s) of the manufacturer(s) of the chemiluminescent kit and anti-TAGE antibody.
Response: According to your suggestion, I have inserted the source details of the relevant materials in Sections 3.3 and 3.4 as follows:
- HRP marker (Cat. no.: BNP-M41; Bionexus, Oakland, CA, USA) solution (Lines 209–210)
- Anti-TAGE antibody and TAGE-BSA (obtained from Prof. Takeuchi of the Department of Advanced Medicine, Medical Research Institute, Kanazawa Medical University, Uchinada, Japan) (Lines 211–213)
- The secondary antibody (Cat. no.: P0448; Dako, Glostrup, Denmark) (Line 232)
- Chemiluminescent kit reagents (Cat. no.:292-69903; Fujifilm Wako Pure Chemical) (Line 234)
- Chemilumi imager (Cat. no.: Fusion FX; M&S Instruments Inc., Osaka, Japan) (Lines 237–238)
Comment 9: Lines 228-229: Has only one component (A right - B right) been multiplied by the calibration corrected value? Or did it apply to both components?
Response: Only one component (ARight – BlankRight) was multiplied with the calibration-corrected value. I have added this information in Lines 254–260.
Comment 10: Table 5: rat primary cardiomyocytes - Why does 4 mM, 24 h treatment appear twice in the list? In addition, the TAGE values given for them are different, although the data come from one reference [24].
Response: In the previous investigation [24], I performed two examinations.
(i) Rat cardiomyocytes were treated with 0, 1, 2, and 4 mM glyceraldehyde for 24 h.
(ii) Rat cardiomyocytes were treated with 4 mM glyceraldehyde for 0, 3, 6, 12, and 24 h.
Therefore, I mentioned both experiments about intracellular TAGE, in which cardiomyocytes were treated with 4mM glyceraldehyde for 24 h. However, I agree with the reviewer that mentioning this information twice may confuse the readers. Therefore, I have deleted one mention of 4 mM glyceraldehyde treatment for 24 h. Please review the revised Table 5, on Line 332.
Comment 11: Table 5: Hep3B - Information on 4 mM, 24 h treatment should be placed at the end of the list.
Response: I agree with the reviewer’s comment and have shifted the relevant information to the end of the list (Table 5, on Line 332).
Comment 12: Figure 7 legend: Last two sentences to be deleted.
Response: I have deleted the last two sentences in the legend of Figure 7.
Comment 13: Lines 602-604: I suggest you use the term "glucose" instead of "glycose". Especially since this term appears more frequently in the manuscript.
Response: I have corrected the word from “glycose” to “glucose” at all instances in the paragraph (Lines 641, 644, 645).

Reviewer 4 Report
The review work of Takanobu Takata aims to describe the new slot blot analysis method for quantifying two types of AGEs. The work is well articulated because it describes in detail the characteristics of the method and highlights its relative advantages compared to the other analytical techniques used previously. Furthermore, the various experimental passages are reported, thus also guaranteeing a possible application to the expert reader.
I believe that the work is well presented, well written and above all able to provide useful information to the scientific sector of interest. In conclusion, it is my opinion that the review can be accepted in its present form.
Author Response
Response Letter to Reviewers’ Comments
Dear Reviewer 4,
Thank you for giving me the opportunity to submit a revised draft of my manuscript titled “Is the Novel Slot Blot a Useful Method for Quantification of Intracellular Advanced Glycation End-Products?” to the journal Metabolites (manuscript ID: 2306811). I appreciate the time and effort that the reviewers have dedicated to providing their valuable feedback on my manuscript. I am grateful to the reviewers for their thoughtful suggestions and insightful comments on my paper, which have enriched the manuscript and produced a better and more balanced account of the research.
I have incorporated changes in the manuscript to reflect the suggestions provided by the reviewers. For the reviewers’ ease, I used yellow highlights to indicate the revised sentences in the manuscript. I also changed the font color of my responses below to red.
Please note that there were two numerical errors in Table 5 “Competition between cell viability and intracellular TAGE” in the first row. These errors have been rectified and highlighted in yellow color. I also inserted a new Ref. 2 and renumbered the previous Ref. 2 to 109, Ref. 109 to 110, and Ref. 110 to 111.
Here is a point-by-point response to the reviewers’ comments and concerns.
Comments of Reviewer 4:
Comments and Suggestions for Authors
The review work of Takanobu Takata aims to describe the new slot blot analysis method for quantifying two types of AGEs. The work is well articulated because it describes in detail the characteristics of the method and highlights its relative advantages compared to the other analytical techniques used previously. Furthermore, the various experimental passages are reported, thus also guaranteeing a possible application to the expert reader.
I believe that the work is well presented, well written and above all able to provide useful information to the scientific sector of interest. In conclusion, it is my opinion that the review can be accepted in its present form.
Response: Thank you for your evaluation against my Review article. Since I rewrote my manuscript based on other Reviewers comments, I believe that the revised manuscript will be accepted for you (I consider that the quality of the revised manuscript will be better than the previous manuscript).

Round 2
Reviewer 3 Report
The author responded to all my comments mentioned in my review and brought to the manuscript the necessary changes that would definitely enhance its quality.